# Synergistic Photocatalytic Oxidation and Reductive Activation of Peroxymonosulfate by Bi-Based Heterojunction for Highly Efficient Organic Pollutant Degradation

**DOI:** 10.3390/nano15060471

**Published:** 2025-03-20

**Authors:** Xiaopeng Zhao, Yang Wang, Fangning Liu, Xiaobin Ye, Shangxiong Wei, Yilin Sun, Jinghui He

**Affiliations:** 1College of Chemistry, Chemical Engineering and Materials Science, Soochow University, Suzhou 215123, China; 19962511733@163.com (X.Z.); wy18735094769@163.com (Y.W.); y15372290159@163.com (X.Y.); weishangxiong2002@163.com (S.W.); lancelott99@163.com (Y.S.); 2State Key Laboratory of Silicate Materials for Architectures, Wuhan University of Technology, Wuhan 430070, China; 3Advanced Water Technology Laboratory, National University of Singapore (Suzhou) Research Institute, Suzhou 215123, China

**Keywords:** photocatalysis, peroxymonosulfate, S-type heterojunction, built-in electric field, ciprofloxacin

## Abstract

Organic pollutants present a substantial risk to both ecological systems and human well-being. Activation of peroxymonosulfate (PMS) have emerged as an effective strategy for the degradation of organic pollutants. Bi-based heterojunction is commonly used as a photocatalyst for reductively activating PMS, but single-component Bi-based heterojunction frequently underperforms due to its restricted absorption spectrum and rapid combination of photogenerated electron–hole pairs. Herein, BiVO_4_ was selected as the oxidative semiconductor to form an S-type heterojunction with CuBi_2_O_4_—x-CuBi_2_O_4_/BiVO_4_ (x = 0.2, 0.5, and 0.8) for PMS photoactivation. The built-in electric field (BEF) in x-CuBi_2_O_4_/BiVO_4_ promoted electron transfer to effectively activate PMS. The x-CuBi_2_O_4_/BiVO_4_ heterojunctions also demonstrate stronger adsorption of the polar PMS than pure CuBi_2_O_4_ or BiVO_4_. In addition, the BEF prompts photoelectrons able to reduce O_2_ to •O_2_^−^ and photogenerated holes in the valence band of BiVO_4_ able to oxidize H_2_O to generate •OH. Therefore, under visible light irradiation, 95.1% of ciprofloxacin (CIP) can be degraded. The 0.5-CuBi_2_O_4_/BiVO_4_ demonstrated the best degradation efficiency and excellent stability in cyclic tests, as well as a broad applicability in degrading other common pollutants. The present work demonstrates the high-efficiency S-type heterojunctions in the coupled photocatalytic and PMS activation technology.

## 1. Introduction

Organic pollutants are resistant to natural degradation and pose a substantial threat to both the environment and mankind [1,2]. This phenomenon results in their pervasive accumulation in natural water bodies [3]. For the past few years, advanced oxidation processes (AOPs) using peroxymonosulfate (PMS) have emerged as an effective strategy to decompose persistent organic pollutants [4] because PMS produces the sulfate radical (SO_4_•^−^) with a long half-life, high redox potential, and broad pH adaptability [5,6]. Traditional activation methods for PMS include ultrasonication [7], light irradiation [8], electrolysis [9], pyrolysis [10], and the use of homogeneous and heterogeneous catalysts [11]. Among them, photoactivation attracts considerable interest due to its environmental benignity, low energy consumption, and high efficiency [12,13]. The crux of photoactivation lies in the ability of the photocatalysts’ conduction bands to provide photoexcited electrons with sufficiently high redox potentials [14] to cleave the peroxide bond of PMS molecules, thereby generating SO_4_•^−^ and •OH radicals (SO_4_•^−^, E^0^ = 2.5–3.1 V [15]; •OH, E^0^ = 1.9–2.8 V [16]).

CuBi_2_O_4_, as a p-type semiconductor owning a narrow bandgap (1.4 eV–1.8 eV) [17], is capable of activating PMS upon visible light illumination through its photoelectrons. The divalent copper ions in CuBi_2_O_4_ may undergo a Cu^2+^/Cu^+^ transition, facilitating electron transfer to PMS for activation. In addition, CuBi_2_O_4_ has been reported in the context of organic photodegradation, excellent reducing properties, low toxicity, high chemical stability, and thermal stability. However, single-component CuBi_2_O_4_ often falls short due to their limited light absorption ranges and rapid recombination of photogenerated electron–hole pairs [18,19]. A promising solution is to construct S-scheme heterojunctions by combining CuBi_2_O_4_ with another semiconductor, where CuBi_2_O_4_ serves as the reductive component and the other as the oxidative component. Such S-scheme heterojunctions are believed to effectively separate photogenerated electron–hole pairs [20,21,22]. The utilization of photogenerated electrons by PMS [17] is found to promote the further separation of photoexcited holes at the oxidative end. If these holes own adequate energy, they can oxidize water to yield •OH radicals, forming an additional photocatalytic oxidation pathway and significantly enhancing the overall degradation performance. Recently, it has been discovered that the built-in electric field (BEF) in S-type heterojunctions can preferentially adsorb polar ions, accelerating mass transfer and further enhancing degradation efficiency [23,24].

In this study, the oxidative n-type semiconductor BiVO_4_ was selected to form an S-type heterojunction with CuBi_2_O_4_ for PMS photoactivation. BiVO_4_ has garnered attention due to its low toxicity, high chemical stability, and thermal stability. The conduction band minimum of BiVO_4_ is approximately −0.11 V (vs. NHE), which enables the photogenerated electrons in BiVO_4_ to react with the photogenerated holes in CuBi_2_O_4_. A series of x-CuBi_2_O_4_/BiVO_4_ (x = 0.2, 0.5, and 0.8) heterojunctions were synthesized via the hydrothermal method. On the one hand, the BEF has facilitated the transfer of electrons, endowing the 0.5-CuBi_2_O_4_/BiVO_4_ heterojunction with superior photocatalytic performance. On the other hand, it boasts the highest PMS adsorption capacity. Using ciprofloxacin (CIP) as a model pollutant, we systematically investigated the key factors influencing the degradation efficiency. Under visible light irradiation, the 0.5-CuBi_2_O_4_/BiVO_4_ can achieve a remarkable 95.1% degradation of CIP into CO_2_ and H_2_O. Moreover, 0.5-CuBi_2_O_4_/BiVO_4_ demonstrated excellent stability in cyclic tests and a broad applicability in degrading other common pollutants, such as rhodamine B, methyl blue, norfloxacin, ofloxacin, oxytetracycline, and tetracycline. This work not only expands the repertoire of photocatalytic degradation strategies for PMS but also provides novel insights into the design of high-performance bismuth-based heterojunctions.

## 2. Materials and Methods

### 2.1. Synthesis of BiVO_4_

Bi(NO_3_)_3_·5H_2_O (5.82 g) and NH_4_VO_3_ (1.40 g) were dissolved in 100 mL of 2 M nitric acid and vigorously stirred at room temperature for 30 min. The pH of the above solution was adjusted to 2 by adding a suitable amount of ammonia, followed by stirring at room temperature for 2 h. The resulting suspension was transferred to a hydrothermal autoclave reactor and kept at 200 °C for 12 h. In the end, the acquired bright yellow powder was repeatedly washed using deionized water and ethanol, followed by vacuum drying at 60 °C.

### 2.2. Synthesis of CuBi_2_O_4_

First, Bi(NO_3_)_3_·5H_2_O (5 mmol) was dissolved in 60 mL of deionized water. Then, Cu(NO_3_)_2_·3H_2_O (2.5 mmol) was added under vigorous stirring. NaOH solution (20 mL, 1 mol/L) was added to the above mixture. The obtained mixture was continuously stirred for 3 h and then positioned in a 100 mL high-pressure reactor for hydrothermal reaction at 180 °C for 18 h. Finally, the obtained product was washed several times using deionized water and ethanol, followed by drying in vacuum at 60 °C.

### 2.3. Synthesis of CuBi_2_O_4_/BiVO_4_ Heterojunctions

Bi(NO_3_)_3_·5H_2_O (242.5 mg) and BiVO_4_ (405 mg) were dispersed in a beaker containing 30 mL of water and stirred for 30 min. Then, Cu(NO_3_)_2_·3H_2_O (60.4 mg) was added to the suspension and stirred for another 30 min. After that, 1 M NaOH (2 mL) was added and continuously stirred for 3 h. Finally, the obtained suspension was transferred to a 50 mL hydrothermal autoclave reactor and reacted at 180 °C for 18 h. After washing and drying, the obtained product was denoted as 0.2-CuBi_2_O_4_/BiVO_4_. Under other unchanged conditions, when the masses of Cu(NO_3_)_2_·3H_2_O and Bi(NO_3_)_3_·5H_2_O were changed to 151.0 mg and 606.2 mg, respectively, the product was denoted as 0.5-CuBi_2_O_4_/BiVO_4_. When the masses of Cu(NO_3_)_2_·3H_2_O and Bi(NO_3_)_3_·5H_2_O were changed to 241.6 mg and 970.1 mg, respectively, the product was denoted as 0.8-CuBi_2_O_4_/BiVO_4_.

### 2.4. CuBi_2_O_4_/BiVO_4_ Performance Evaluation Tests

The degradation of CIP was carried out using a customized photocatalytic reaction device (CEAuLight, CEL-PF300-T9, Beijing, China) to evaluate its catalytic performance. The light source equipped in this device is a 420 W xenon lamp with a 300 nm filter. At the same time, a circulating cooling device is used to control the temperature to always remain at 25 °C. Usually, after adding CuBi_2_O_4_/BiVO_4_ to a CIP solution with a certain concentration (50 mL), the mixture is fully stirred under dark conditions to completely eliminate the influence of adsorption. After sufficient stirring, a certain amount of PMS was added while switching on the light source. After reacting for a certain period of time, the mixed solution was filtered using a 0.22 μm polyethersulfone membrane needle filter and then quenched by adding methanol solution immediately. Finally, the concentration of CIP was assessed by the absorbance of the filtrate at 276 nm.

Additionally, a quasi-primary reaction kinetics model (Equation (1)) was used to verify the degradation process:ln(C_0_/C_t_) = k_t_(1)
where t represents the reaction time, C_0_ represents the original concentration of CIP, C_t_ refers to the concentration of CIP at time t, and k is the apparent rate constant.

## 3. Results

### 3.1. Structure and Morphology Analysis

Figure 1a shows the in situ procedure of CuBi_2_O_4_ in BiVO_4_. The crystal structures of the synthesized catalysts were characterized by X-ray diffraction (XRD) and are shown in Figure 1b. The BiVO_4_ sample has major peaks matching very well with the standard card of monoclinic BiVO_4_ (JCPDS 14-0688, Suzhou, China) [25], among which 18.6°, 18.9°, 28.9°, 30.5°, 34.5°, 35.2°, 39.8°, 42.4°, 46.7°, 47.3°, 50.3°, 53.3°, 58.4°, and 59.4° correspond, respectively, to the diffraction characteristic peaks of the (110), (011), (121), (040), (200), (002), (211), (240), (042), (202), (161), (321), and (123) crystal planes of monoclinic BiVO_4_, which confirms the successful preparation of pure BiVO_4_. For pure CuBi_2_O_4_, three distinct peaks at 2θ = 21.0°, 28.2°, and 37.6° are attributable to the (200), (131), and (151) crystal planes of CuBi_2_O_4_. The diffraction features of both CuBi_2_O_4_ and BiVO_4_ co-occur in the CuBi_2_O_4_/BiVO_4_ samples, proving the successful synthesis of the heterojunction. As the content of CuBi_2_O_4_ increases, the peak intensity of BiVO_4_ in the (CuBi_2_O_4_/BiVO_4_) composites decreases accordingly, but no diffraction peaks for other impurities are found.

Fourier transform infrared spectroscopy (FT-IR) and scanning electron microscopy (SEM) were used to further characterize the structure of these samples (Figure 1c). For the original CuBi_2_O_4_, the peak at 859 cm^−1^ and 1398 cm^−1^ are related to the stretching vibration of the Bi–O bond [26,27]. The peak at 556 cm^−1^ is related to the stretching vibration of the Cu–O bond [28]. The broad peak at 742 cm^−1^ represents the VO_4_^3−^ antisymmetric stretching vibration of BiVO_4_ [29]. These representative peaks of CuBi_2_O_4_ and BiVO_4_ can also be detected in the CuBi_2_O_4_/BiVO_4_ complex. The original BiVO_4_ exhibited an irregular blocky structure (Appendix A), while CuBi_2_O_4_ showed a structure similar to a rod-like one (Appendix A). In addition, Appendix A and Figure 1d present the morphological structures of the 0.2-CuBi_2_O_4_/BiVO_4_, 0.8-CuBi_2_O_4_/BiVO_4_, and 0.5-CuBi_2_O_4_/BiVO_4_ composites. It can be clearly observed that on the basis of retaining the original BiVO_4_ structure, CuBi_2_O_4_ grows uniformly on the surface of BiVO_4_. It is worth noting that as the loading amount of CuBi_2_O_4_ increases, the roughness of the composite surface also increases accordingly, which indicates that the CuBi_2_O_4_ is in close contact with the BiVO_4_ surface. The elemental analysis (EDX) illustrates the presence of Cu, Bi, O, and V elements in 0.5-CuBi_2_O_4_/BiVO_4_ (Figure 1e). All of these pieces of evidence prove the successful combination of the two components in CuBi_2_O_4_/BiVO_4_. The morphology and composition of 0.5-CuBi_2_O_4_/BiVO_4_ were further observed using a high-resolution transmission electron microscope (HRTEM). In Figure 1f, two different lattice fringes are clearly shown. The dark area represents CuBi_2_O_4_ with a lattice spacing of 0.318 nm, assigned to the (211) crystal plane of CuBi_2_O_4_ [26]; and the bright area represents BiVO_4_ with a lattice spacing of 0.308 nm, corresponding to the (112) crystal plane of BiVO_4_ [30]. The close contact between the CuBi_2_O_4_ and BiVO_4_ crystals can be clearly seen in the Figure 1f. Moreover, in Figure 1g, the grain boundaries between CuBi_2_O_4_ and BiVO_4_ may be interconnected via Bi-O-V bonds. This structural arrangement endows the interface with enhanced electronic transport properties and stability, significantly outperforming those of heterojunctions formed through mere physical mixing [31].

To investigate the chemical valence states changes and electron transfer of BiVO_4_ and CuBi_2_O_4_ before/after forming heterojunctions, we selected 0.5-CuBi_2_O_4_/BiVO_4_ for x-ray photoelectron spectroscopy (XPS). In Figure 2a, the Bi 4f spectrum of CuBi_2_O_4_ shows two peaks at 163.98 and 158.68 eV, which are Bi 4f_5/2_ and Bi 4f_7/2_, respectively [32]. For BiVO_4_, the two peaks at 164.33 and 159.03 eV can also be attributed to Bi 4f_5/2_ and Bi 4f_7/2_ [33], which proves the existence of Bi^3+^ in CuBi_2_O_4_ and BiVO_4_. For the 0.5-CuBi_2_O_4_/BiVO_4_ composite, the Bi 4f peak is located between the peaks of BiVO_4_ and CuBi_2_O_4_. Similarly, the O1s peak of the composite is also located between CuBi_2_O_4_ and BiVO_4_, which implies a strong interaction between BiVO_4_ and CuBi_2_O_4_ (Figure 2b). For CuBi_2_O_4_, the two peaks near 953.74 and 933.88 eV can be labeled as Cu 2p_1/2_ and Cu 2p_3/2_, respectively [34] (Figure 2c). The satellite peaks of copper at 943.60 and 940.51 eV indicate the existence of Cu^2+^ in the CuBi_2_O_4_ sample [35]. After being combined with BiVO_4_, the Cu 2p peak shifts to a higher binding energy position (953.89 and 934.03 eV), which indicates that electron transfer from CuBi_2_O_4_ to BiVO_4_ occurs in the 0.5-CuBi_2_O_4_/BiVO_4_ heterostructure. For the V 2p spectrum of pure BiVO_4_ (Figure 2d), the two typical peaks at 524.31 eV and 516.77 eV are attributed to V 2p_1/2_ and V 2p_3/2_, respectively, which matches with V^5+^ ions [25]. When BiVO_4_ and CuBi_2_O_4_ are in contact, the binding energy of the two characteristic peaks of V 2p shifts in a smaller direction (524.17 eV and 516.63 eV), which indicates that BiVO_4_ accepts electrons from CuBi_2_O_4_. In conclusion, the results of XPS prove the formation of the CuBi_2_O_4_/BiVO_4_ heterostructure and the strong interaction between the heterostructures, which may contribute to the migration of photogenerated electrons and holes and thereby improve the photocatalytic activity.

### 3.2. Analysis of Photocatalytic Performance

Further photoelectron properties of the catalysts are investigated by ultraviolet–visible (UV) spectroscopy. As shown in Appendix A, the original BiVO_4_ and CuBi_2_O_4_ show absorption edges at approximately 550 nm and 800 nm, respectively. CuBi_2_O_4_ clearly exhibits extensive absorption of visible light. When BiVO_4_ and CuBi_2_O_4_ are combined, the absorption edge of CuBi_2_O_4_/BiVO_4_ shows a significant redshift compared to the original BiVO_4_, indicating that the optical absorption performance is further improved compared to BiVO_4_. The band gaps of BiVO_4_ and CuBi_2_O_4_, calculated by the Tacu equation, are 2.41 eV and 1.73 eV, respectively (Appendix A).

Through the Mott–Schottky curves in Appendix A, the band edge positions of the two semiconductors were evaluated. The flat band positions of BiVO_4_ and CuBi_2_O_4_ are 0.11 V and 1.14 V, respectively, relative to the normal hydrogen electrode (NHE). In addition, the slope of the Mott–Schottky curves of CuBi_2_O_4_ and BiVO_4_ indicates that they are p-type and n-type semiconductors, respectively. Founded on the equation E_VB_ = E_CB_ + Eg, the valence band potential of BiVO_4_ and the conduction band potential of CuBi_2_O_4_ are obtained as 2.30 V and −0.59 V (vs. NHE), respectively. As shown in Figure 3d, there is an interleaved distribution between the energy bands of BiVO_4_ and CuBi_2_O_4_. Therefore, it is reasonable to speculate that there may be two charge transfer modes (type-II and S-type schemes) in the 0.5-CuBi_2_O_4_/BiVO_4_ heterojunction.

We must first rule out that CuBi_2_O_4_/BiVO_4_ belongs to the type-II heterojunction. Taking 0.5-CuBi_2_O_4_/BiVO_4_ as the typical heterojunction, both BiVO_4_ and CuBi_2_O_4_ can be excited by visible light irradiation to generate electrons and holes (Equations (2) and (3). As in the case of type-II heterojunction, the photogenerated electrons in CuBi_2_O_4_ would transfer to the BiVO_4_ conduction band. At the same time, the holes generated in BiVO_4_ would migrate to the CuBi_2_O_4_ valence band. The reduction potential of (O_2_/•O_2_^−^) is −0.32 V (vs. NHE), which is more negative than the conduction band potential of BiVO_4_ (−0.11 V vs. NHE) [36]; the oxidation potential of (•OH/OH^−^) is 2.27 V (vs. NHE), which is more positive than the valence band potential of CuBi_2_O_4_ (1.14 V vs. NHE) [37]. Thus, once the heterojunction belongs to the type-II heterojunction, •O_2_^−^ and •OH will not be generated. In fact, this assumption is negated because there are distinct •O_2_^−^ and •OH peaks in the electron paramagnetic resonance spectroscopy (EPR) test.

Different from the type-II heterojunction, electrons transfer from the surface of CuBi_2_O_4_ to BiVO_4_, forming a built-in-electric field (BEF) pointing from CuBi_2_O_4_ to BiVO_4_. Under the action of such an electric field, the band edge of CuBi_2_O_4_ will bend upward due to the loss of electrons; and the band edge of BiVO_4_ will bend downward as a result of electron accumulation. From the above conclusions, it can be confirmed that the 0.5-CuBi_2_O_4_/BiVO_4_ heterojunction belongs to the S-type heterojunction. In addition, the electrons remaining at the conduction band of CuBi_2_O_4_ have sufficient potential to reduce O_2_ to generate •O_2_^−^ (Equation (5)), and the holes in the valence band of BiVO_4_ have a sufficient positive potential to oxidize H_2_O to generate •OH (Equation (6)). Therefore, the S-type heterojunction not only promotes the separation of photogenerated carriers but also maintains high redox activity, thereby synergistically enhancing the catalytic activity. Additionally, the photoexcited electrons remaining on CuBi_2_O_4_ can be arrested by PMS to yield SO_4_•^−^ (Equation (7)); then, a part of SO_4_•^−^ may react with H_2_O to generate •OH (Equation (8)) [38]. Thus, the incorporation of PMS utilizes more photocarriers and boosts the catalytic efficiency, encompassing both the removal rate and mineralization rate of CIP.

To further investigate the effect of light on the chemical valence state changes and electron transfer of the 0.5-CuBi_2_O_4_/BiVO_4_ S-scheme heterojunction, the in situ XPS was obtained. After illumination, the binding energy of V 2p increased by 0.1 eV, while the binding energy of Cu 2p decreased, indicating electron transfer from V to Cu (Figure 3a,b). Moreover, according to the Auger electron spectrum (AES), a small amount of Cu^2+^ is reduced to Cu^+^, which is consistent with the electron transfer situation (Figure 3c). Combining the analysis of energy bands, when CuBi_2_O_4_ and BiVO_4_ are in contact, the interface charge will flow into the BiVO_4_ and create a BEF at the 0.5-CuBi_2_O_4_/BiVO_4_ interface. Under illumination, the electrons of BiVO_4_ are excited to transition to the VB and flow towards CuBi_2_O_4_ under the promotion of BEF (Figure 3d) [20].

The XPS spectra of the catalyst both before and after the reaction also confirm the valance change of Cu. As shown in Appendix A, the Cu^+^ species obviously appears in 0.5-CuBi_2_O_4_/BiVO_4_ after the reaction, while the other metal elements in the catalyst do not show significant changes in composition and valence state. Therefore, it can be speculated that in the catalytic stage, Cu^2+^ is partially converted to Cu^+^ on the catalyst surface, which proves that the Cu^+^/Cu^2+^ redox pair participates in the catalytic system and can activate PMS to generate various reactive substances. In addition, the XPS of Bi 4f, V 2p, and O 1s do not show significant changes before and after the reaction, indicating that the 0.5-CuBi_2_O_4_/BiVO_4_ catalyst possesses reliable stability, which provides a potential possibility for the practical application of this strategy.

The photocurrent responses of a series of prepared catalysts under visible light are shown in Figure 3e. The photocurrent intensities decay as the following trend: 0.5-CuBi_2_O_4_/BiVO_4_ > 0.8-CuBi_2_O_4_/BiVO_4_ > 0.2-CuBi_2_O_4_/BiVO_4_ > BiVO_4_ > CuBi_2_O_4_. A steady-state fluorescence spectroscopy (PL) test was conducted (in Figure 3f); bare CuBi_2_O_4_ and BiVO_4_ exhibit strong emission signals due to the severe recombination of their photoexcited charge carriers. In contrast, the peak signal of the CuBi_2_O_4_/BiVO_4_ heterostructure is significantly reduced. Additionally, the 0.5-CuBi_2_O_4_/BiVO_4_ catalyst exhibits the lowest PL intensity, indicating that the recombination of photoexcited electron–hole pairs is most effectively suppressed. Hence, the construction of the CuBi_2_O_4_/BiVO_4_ heterostructure can significantly suppress charge recombination and prompt the performance of photoactivated PMS. The transient photocurrent response curve and the electrochemical impedance spectroscopy (EIS) curve can also illustrate the carrier dynamics behavior. In Figure 3g, it can be distinctly observed that the radius corresponding to 0.5-CuBi_2_O_4_/BiVO_4_ is the smallest, which is also in line with the above test results, demonstrating that the heterostructure has the best interfacial carrier migration ability. Appendix A illustrates the equivalent circuit diagram employed for the EIS analysis of 0.5-CuBi_2_O_4_/BiVO_4_, and Appendix A summarizes the corresponding fitting parameters.

### 3.3. PMS Adsorption Test

The S-type heterojunctions not only improve the photoelectronic properties but also enhance the PMS adsorptions. The content of PMS in the solution was determined by the colorimetric method. As shown in Figure 4a, when PMS, 0.5-CuBi_2_O_4_/BiVO_4_, Co^2+^, and TMB coexisted simultaneously, the solution changed from colorless to blue, and an oxidation peak of TMB appeared at 652 nm in the UV spectrum. This is due to the fact that the free radicals (SO_4_•^−^) generated by the decomposition of PMS and Co²⁺ through the single-electron transfer pathway oxidized TMB to form blue TMB_ox_ [39]. However, when the content of PMS was excessive or the reaction time was too long, TMB was over-oxidized to form a two-electron oxidation product (TMB_ox2_) [40]. When a new oxidation peak appeared at 452 nm, the absorption peak at 652 nm decreased significantly (Appendix A). In order to avoid the impact of the two-electron oxidation product on the experimental results, we strictly controlled the reaction time (15 min), temperature (37 °C), and the content of PMS in the solution to be tested (<0.3 mM). Eventually, accurate measurement of PMS was achieved. Based on the above conditions, a standard curve for detecting PMS was constructed (Figure 4b). When the concentration of PMS ranged from 0 to 0.3 mM, there was a good linear relationship between the concentration of PMS and absorbance, and the detection limit was as low as 2.918 μM (Figure 4c). This result laid a foundation for accurately measuring the content of PMS in the solution.

Compared to CuBi_2_O_4_ and CuBi_2_O_4_ existing alone, due to the integration of the two, the specific surface area of the heterojunction formed by them, i.e., 0.5-CuBi_2_O_4_/BiVO_4_, has been improved slightly (Appendix A) [41,42,43,44]. This helps to expose more active sites and adsorption sites, enhance the affinity between the materials and PMS, and thus improve the adsorption and activation effects of 0.5-CuBi_2_O_4_/BiVO_4_ on PMS. [43]. In this study, the isotherm of the Langmuir theoretical model was adopted to investigate the adsorption behaviors of BiVO_4_, CuBi_2_O_4_, and 0.5-CuBi_2_O_4_/BiVO_4_ towards PMS (Appendix A). The adsorption kinetic curves indicate that all three materials exhibited rapid and maximum adsorption of PMS within 20 min and reached equilibrium after 30 min. This rapid kinetic behavior suggests that the adsorption process depends on the binding sites available on the materials for PMS adsorption. After 30 min, the surface of the adsorbing material becomes saturated with the PMS and finally reaches a dynamic equilibrium (Appendix A). Moreover, the initial concentration of PMS also influences the adsorption performance of the adsorbing materials. As shown in Appendix A, with the increase in the amount of PMS, the adsorption amounts of the materials for PMS increased to varying degrees. Although the K_L_ value of 0.5-CuBi_2_O_4_/BiVO_4_ (0.577 g·mmol^−1^·min^−1^) is slightly lower than that of BiVO_4_ (0.623 g·mmol^−1^·min^−1^), its larger specific surface area provides more adsorption sites, enabling its maximum adsorption amount for PMS to reach 0.804 mmol·g⁻^1^, which is higher than the adsorption amounts of BiVO_4_ (0.360 mmol·g^−1^) and CuBi_2_O_4_ (0.212 mmol·g⁻^1^) when they exist alone. The above results demonstrate that compared with BiVO_4_ and CuBi_2_O_4_ alone, 0.5-CuBi_2_O_4_/BiVO_4_ exhibits a higher affinity for PMS and excellent adsorption performance. The effective adsorption of PMS provides favorable conditions for catalytic reactions.

### 3.4. Analysis of PMS Activation Mechanism

In order to clarify the catalytic degradation mechanism, the active species in the 0.5-CuBi_2_O_4_/BiVO_4_/vis/PMS system were determined through free radical quenching experiments. Usually, methanol (MA) can be used to quench •OH and SO_4_•^−^, while tert-butanol (TBA) can only quench •OH. In addition, dimethyl sulfoxide (DMSO), triethanolamine (TEA), p-benzoquinone (p-BQ), and L-histidine (L-histidine) were selected as the quenchers for e^−^, h^+^, •O_2_^−^, and ^1^O_2_, respectively. As shown in Figure 5a, when TBA and MeOH were added into the 0.5-CuBi_2_O_4_/BiVO_4_/vis/PMS system, the removal rate of CIP decreased from 95.1% to 70.3% and 43.2%, respectively. This suggests that both •OH and SO_4_•^−^ are involved in CIP degradation. In addition, when a certain amount of DMSO, p-BQ, TEA, and L-histidine were added to the system, respectively, the degradation efficiency of CIP was affected to varying degrees. So, it can be concluded that •O_2_^−^, h^+^, ^1^O_2_, e^−^, SO_4_•^−^, and •OH all contribute to the degrading of CIP. In addition, EPR spectroscopy was utilized to further corroborate the active species that were generated during the catalytic reaction phase. It is evident that no signal of any spin adduct is detected under dark conditions (Figure 5b–d), suggesting that the catalyst itself was unable to generate active substances. However, when under light conditions, the quadruple signal peak of DMPO-•OH (Figure 5b), the sextuple signal peak of DMPO-•O_2_^−^ (Figure 5c), and the triple signal peak of TEMP-^1^O_2_ (Figure 5d) were observed, which indicates that 0.5-CuBi_2_O_4_/BiVO_4_ can generate •O_2_^−^, •OH, and ^1^O_2_ under visible light. More notably, when light and PMS exist simultaneously, the adduct signal intensity of the system is significantly higher than that of the light condition or PMS alone. This suggests a pronounced synergistic effect between photocatalysis and PMS activation.

In addition, the EPR test results show that there is non-radical ^1^O_2_ in the 0.5-CuBi_2_O_4_/BiVO_4_/vis/PMS system. In fact, ^1^O_2_ can be generated through the following approach: (i) ^1^O_2_ can be obtained by the reaction of •O_2_^−^ with h^+^, H_2_O, and •OH (Equations (13)–(15)) [45,46]; (ii) the weakly oxidizing SO_5_•^−^ may react with H_2_O to generate ^1^O_2_ (Equation (16)) [47]; (iii) ^1^O_2_ can also be generated via PMS autolytic decomposition (Equation (17)) [48]. In summary, the high efficiency charge separation efficiency of the S-type heterojunction photocatalyst not only provides an abundance of photogenerated electrons for the activation of PMS but also facilitates the Cu^+^/Cu^2+^ redox cycle, thereby further enhancing the rapid activation of PMS. Meanwhile, the redox couple and PMS can consume ample photogenerated electrons. The aforementioned concerted action between PMS activation and photocatalysis is beneficial for generating a greater amount of active species. Ultimately, under the influence of this synergistic effect of h^+^, •O_2_^−^, •OH, SO_4_•^−^, and ^1^O_2_, CIP is degraded into smaller molecules and inorganic ions in the 0.5-CuBi_2_O_4_/BiVO_4_/vis/PMS catalytic system.
BiVO_4_ + hν → e^−^ (BiVO_4_) + h^+^ (BiVO_4_)(2)
CuBi_2_O_4_ + hν → e^−^ (CuBi_2_O_4_) + h^+^ (CuBi_2_O_4_)(3)
e^−^ (BiVO_4_) + h^+^ (CuBi_2_O_4_) → Recombination(4)
e^−^ (CuBi_2_O_4_) + O_2_ → •O_2_^−^(5)
h^+^ (BiVO_4_) + H_2_O → •OH+H^+^(6)
HSO_5_^−^ + e^−^ → SO_4_•^−^ + OH^−^(7)
SO_4_•^−^ + H_2_O→SO_4_^2−^ + •OH + H^+^(8)
Cu^2+^ + e^−^→Cu^+^(9)
Cu^+^ + HSO_5_^−^ → Cu^2+^ + SO_4_•^−^ + OH^−^(10)
Cu^+^ + HSO_5_^−^ → Cu^2+^ + SO_4_^2−^ + •OH(11)
Cu^2+^ + HSO_5_^−^ → Cu^+^ + SO_5_•^−^ + H^+^(12)
•O_2_^−^ + h^+^(BiVO_4_) → ^1^O_2_(13)
2•O_2_^−^ + 2H_2_O → ^1^O_2_ + H_2_O_2_ + 2OH^−^(14)
•O_2_^−^ + •OH → ^1^O_2_ + OH^−^(15)
SO_5_•^−^ + H_2_O→ ^1^O_2_ + SO_4_^2−^ + 2H^+^(16)
HSO_5_^−^ + SO_5_^2−^ → SO_4_^2−^ + HSO_4_^−^ + ^1^O_2_(17)

### 3.5. Degradation Performance of CIP

Photodegradation experiments of CIP were first carried out in the absence of a catalyst, and the content of CIP remained almost unchanged when only PMS was present in the system, which further proved that the photocatalyst was necessary for the activation of PMS (Appendix A).

In order to explore the optimal conditions of the 0.5-CuBi_2_O_4_/BiVO_4_/vis/PMS system, the impacts of various experimental conditions on the removal of CIP were studied. The effect of the initial CIP concentration on the 0.5-CuBi_2_O_4_/BiVO_4_/vis/PMS system is shown in Appendix A. As the initial concentration increased from 5 to 10 mg/L, the degradation rate of CIP gradually increased from 89.5% to 94.8% within 30 min. Nevertheless, the degradation efficiency of CIP decreased at higher CIP concentrations to 71.6% (15 mg/L) and 54.4% (20 mg/L), respectively. Appendix A shows the catalyst dosage influence on the degradation efficiency in the 0.5-CuBi_2_O_4_/BiVO_4_/vis/PMS system. Specifically, when the catalyst increased from 100 to 400 mg/L, the removal rate of CIP increased from 39.5% to 94.8% within 30 min. It is notable that when the catalyst content continued to increase from 400 mg/L to 500 mg/L, the degradation rate of CIP decreased from 94.8% to 82.7% within 30 min. This phenomenon can be attributed to the increase in the turbidity of the solution due to an excessive amount of catalyst. This, in turn, reduces the light absorption efficiency, leading to a significant decrease in the removal rate of CIP. Consequently, a catalyst dosage of 400 mg/L of 0.5-CuBi_2_O_4_/BiVO_4_ was determined to be optimal for this study.

Similarly, the influence of PMS concentration ranging from 0.25 mmol/L to 1.25 mmol/L on the removal of CIP was studied. Appendix A shows that as the amount of PMS added increases, the catalytic performance of the 0.5-CuBi_2_O_4_/BiVO_4_/vis/PMS system gradually improves. In this system, PMS functions as an electron acceptor. This action facilitates the separation of photogenerated charges within the catalyst, thereby boosting the degradation efficacy. However, when the addition amount of PMS increases from 1 mmol/L (94.8%) to 1.25 mmol/L (86.5%), the removal rate of CIP shows a slight decrease. This result may be attributed to the following three factors: first, a limited quantity of catalyst is incapable of furnishing an adequate number of active sites to fully activate the surplus PMS; second, immoderate PMS may capture SO_4_•^−^, generating less-reactive SO_5_•^−^ free radicals (Equation (18)) [49]; third, SO_4_•^−^ may exhibit a self-quenching effect in systems with high concentrations (Equation (19)) [50]. Therefore, based on the above analysis, adding 1 mmol/L PMS to the 0.5-CuBi_2_O_4_/BiVO_4_/vis/PMS system is the most optimal.
HSO_5_^―^ + SO_4_•^−^ → SO_5_•^−^ + SO_4_^2―^ + H^+^(18)
SO_4_•^−^ + SO_4_•^−^ → S_2_O_8_^2―^(19)

The impact of pH on the eliminate of CIP in the 0.5-CuBi_2_O_4_/BiVO_4_/vis/PMS system was studied (Appendix A). At the beginning of the reaction, the removal of CIP increased with the increase of solution pH. When the pH value gradually changed from 3 to 7, the removal rate of CIP increased from 79.1% to 94.9%. When the pH increased, the CIP removal rate ultimately decreased to 64.4% at PH = 11. These findings suggest that neutral or slightly acidic conditions are conducive to the removal of CIP, whereas strongly acidic or strongly alkaline systems weaken the removal performance. The strong acid system may disrupt the protonation and deprotonation balance of •O_2_^−^, resulting in the consumption of •O_2_^−^; while for the strongly alkaline environment, SO_4_•^−^ tends to transform into •OH with relatively lower reactivity (Equation (20)) [51]. In addition, the highly alkaline environment may induce the deprotonation of PMS, thereby facilitating its transformation into the weakly oxidizing SO_5_^2−^, resulting in a decrease in the removal efficiency of CIP. Within the pH range of 3 to 7, the 0.5-CuBi_2_O_4_/BiVO_4_/vis/PMS still maintains a removal rate of over 80% for CIP, which also indicates that the 0.5-CuBi_2_O_4_/BiVO_4_/vis/PMS system has practical application value in water treatment.
SO_4_•^−^ + OH^―^ → •OH + SO_4_^2―^(20)

The ratio of different components in x-CuBi_2_O_4_/BiVO_4_ is an important factor affecting its CIP degradation performance. Figure 6a illustrates the removal curves of CIP for different catalysts under visible light. Following a 30-min dark adsorption test, all prepared catalysts achieved adsorption–desorption equilibrium, with the final CIP adsorption rate being less than 10% for each. In the presence of the catalyst, the concentration of CIP decreases as the irradiation time increases. Specifically, due to the relatively severe recombination of photo-induced charge carriers, bare BiVO_4_ and CuBi_2_O_4_ only show CIP removal rates of 19.2% and 6.7%, respectively, within 30 min of irradiation. The heterostructure (0.2-CuBi_2_O_4_/BiVO_4_) with a molar ratio of CuBi_2_O_4_ to BiVO_4_ of 0.2:1 increases the photocatalytic degradation rate of CIP to 39.5%. With the further increase in the CuBi_2_O_4_ loading amount, after 30 min of irradiation, 0.5-CuBi_2_O_4_/BiVO_4_ further increases the removal rate of CIP to 60.8%. On the one hand, the loading of CuBi_2_O_4_ can further increase the light absorption ability of BiVO_4_; on the other hand, the close contact between the two components promotes the separation of photogenerated charges, thus enhancing the photocatalytic efficiency. However, as the CuBi_2_O_4_ loading amount further increases (0.8-CuBi_2_O_4_/BiVO_4_), the degradation efficiency of CIP drops to 50.2%, which is due to the excessive CuBi_2_O_4_ not only hindering the light absorption of the catalyst but also possibly covering the active sites, affecting the photocatalytic performance. Figure 6b shows the CIP degradation curve on various catalysts under the presence of PMS. It can be clearly observed that only in the first 5 min, the degradation rate of CIP is about 10%. This indicates that the activation ability of all the prepared photocatalysts for PMS without illumination is extremely limited.

To explore the cooperative interaction between PMS activation and photocatalysis more deeply, CIP was degraded using individual components and x-CuBi_2_O_4_/BiVO_4_ in the vis/PMS system (Figure 6c). Within 30 min, the removal rates of CIP by bare BiVO_4_ and CuBi_2_O_4_ in the presence of light and PMS are 56.5% and 17.4%, respectively. These rates are higher than those achieved by the catalysts under either single visible light or PMS conditions alone. It is worth mentioning that the 0.5-CuBi_2_O_4_/BiVO_4_/vis/PMS system demonstrates the highest CIP removal rate of 95.1% within 30 min. At the same time, the mineralization efficiency of 0.5-CuBi_2_O_4_/BiVO_4_/vis/PMS in degrading CIP was determined. As shown in Figure 6d, e, the total organic carbon (TOC) removal rates of the 0.5-CuBi_2_O_4_/BiVO_4_/PMS, 0.5-CuBi_2_O_4_/BiVO_4_/vis, and 0.5-CuBi_2_O_4_/BiVO_4_/vis/PMS systems within 30 min are 3.8%, 29.8%, and 66.9%, respectively. The test results show that the 0.5-CuBi_2_O_4_/BiVO_4_/vis/PMS system has the best mineralization ability. It can be concluded from the above analysis that there is a synergistic interaction between photocatalysis and PMS activation in the 0.5-CuBi_2_O_4_/BiVO_4_/vis/PMS system, and a large number of active substances may be generated through multiple channels, thereby significantly accelerating the degradation of CIP. In addition, we performed Raman spectroscopy analysis on the used 0.5-CuBi_2_O_4_/BiVO_4_, without washing, to investigate potential residual species. As illustrated in Appendix A, no characteristic peaks corresponding to CIP or PMS were detected on the catalyst surface after the reaction. This observation indicates that the residual CIP did not adsorb onto the 0.5-CuBi_2_O_4_/BiVO_4_, further supporting the reliability and accuracy of the TOC test results. Moreover, compared to fresh catalysts, the characteristic peaks did not change, which also indicates good catalytic stability.

For the purpose of further certifying the universality of the 0.5-CuBi_2_O_4_/BiVO_4_/vis/PMS system, the degradation effects on rhodamine B (RhB), methyl blue (MB), norfloxacin (NOR), ofloxacin (OFL), oxytetracycline (OTC), and tetracycline (TC) were tested, respectively (Figure 6f). Within 30 min, the 0.5-CuBi_2_O_4_/BiVO_4_/vis/PMS system demonstrated remarkable degradation efficiency for a variety of organic pollutants and common antibiotics. In 5 min, the degradation rate for RhB, MB, NOR, OFL, OTC, and TC was 99.7%, 95.7%, 83.0%, 83.5%, 99.3%, and 99.7%, respectively. This fully illustrates that the 0.5-CuBi_2_O_4_/BiVO_4_/vis/PMS system possesses extensive applicability and promising prospects for application.

In the 0.5-CuBi_2_O_4_/BiVO_4_/vis/PMS system, the influence of common anions on the decomposition of CIP is shown in Figure 6g. It can be found that compared with the control group, the introduced anions exert varying levels of impact on the catalytic degradation efficiency of CIP. Especially in the presence of 10 mmol/L HCO_3_^−^ and HPO_4_^2−^, a significant inhibitory effect on the removal of CIP is observed. The reduced CIP removal efficiency may stem from the scavenging behavior of anions on h^+^, SO_4_•^−^, and •OH, resulting in the generation of fewer oxidizing free radicals, and thereby inhibiting the catalytic performance of the 0.5-CuBi_2_O_4_/BiVO_4_/vis/PMS system. The stability of the catalyst is of paramount importance for practical applications. As shown in Figure 6h, after being recycled five times, the removal rate of CIP still remained above 80%, indicating that the 0.5-CuBi_2_O_4_/BiVO_4_ photocatalyst has excellent stability.

### 3.6. Degradation Pathway of CIP

Liquid chromatography–mass spectrometry (LC-MS) technology was used to determine the intermediates generated in the catalytic system (Appendix A). Rooted in the determined results, three degradation paths of CIP were speculated. As shown in Figure 7, the three paths, respectively, go through steps such as dehydroxylation, demethylation, defluorination, hydroxylation, and complete rupture of the piperazine ring structure. Ultimately, under the continuous influence of multiple active species, these decomposition results will be further broken down and mineralized into smaller organic molecules, inorganic ions, CO_2_, and H_2_O.

Since excess CIP dispersed in the environment is potentially toxic to organisms in water bodies, in order to verify the practical significance of our proposed CIP degradation system, we assessed the biotoxicity of the intermediate products formed during the degradation process by testing the inhibitory effect of the solution on the growth of *Escherichia coli* (*E. coli*) before and after the reaction, using *E. coli* as an example. As shown in Appendix A, the initial CIP solution before the reaction had a high inhibition rate, and the cell survival rate of *E. coli* was only 4.2% (Appendix A). After the end of the catalytic degradation process, the cell survival rate of *E. coli* reached 42.3% (Appendix A), which indicates that the removal of CIP reduces its own higher biotoxicity and converts it to a less toxic small molecule.

## 4. Conclusions

In summary, herein, a series of Bi-based heterojunctions, i.e., x-CuBi_2_O_4_/BiVO_4_ (x = 0.2, 0.5, and 0.8), were successfully prepared by a simple hydrothermal method. By controlling the optimal loading amount of CuBi_2_O_4_, 0.5-CuBi_2_O_4_/BiVO_4_ exhibited a higher adsorption and activation capacity for PMS and demonstrated excellent photocatalytic degradation behavior towards CIP under visible light. Characterization tests and mechanism analyses revealed that the enhanced photocatalytic activity of 0.5-CuBi_2_O_4_/BiVO_4_ was attributed to the formation of an S-type heterostructure with a BEF. This structure expedites the separation of photogenerated charge carriers while preserving robust redox capabilities. At the same time, the photogenerated electrons and redox cycling sites (Cu^+^/Cu^2+^) effectively enabled PMS, and the catch of photogenerated electrons by PMS further suppresses the recombination of electron–hole pairs, thereby generating a greater number of active species and remarkably improving the catalytic properties. Additionally, the 0.5-CuBi_2_O_4_/BiVO_4_/vis/PMS system not only exhibits a certain degree of adaptability to various factors such as solution pH and the presence of anions but also demonstrates relatively rapid degradation efficiency towards other waterborne organic pollutants. The present work not only demonstrates the high efficiency of the coupled photocatalytic and PMS activation technology but also has some inspirational significance for practical water treatment technology.

## Figures and Tables

**Figure 1 nanomaterials-15-00471-f001:**
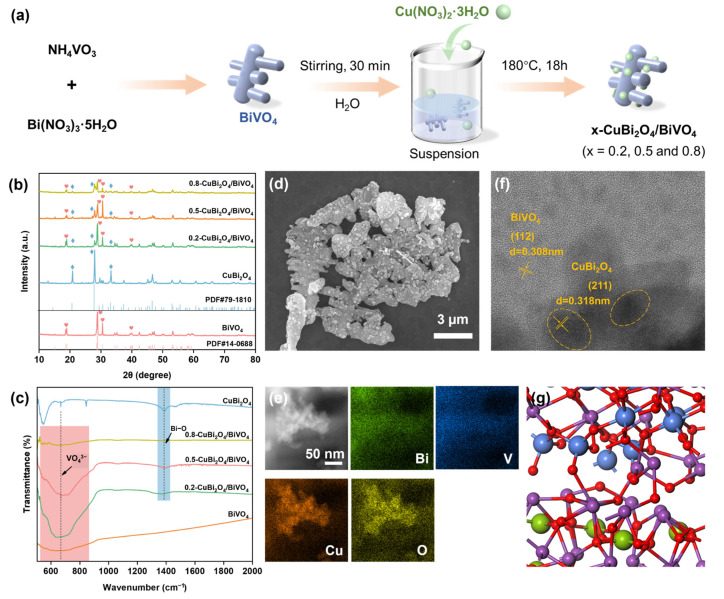
(**a**) The synthesis procedure of x-CuBi_2_O_4_/BiVO_4_. (**b**) The XRD patterns of the catalysts of BiVO_4_, CuBi_2_O_4_, 0.2-CuBi_2_O_4_/BiVO_4_, 0.5-CuBi_2_O_4_/BiVO_4_, and 0.8-CuBi_2_O_4_/BiVO_4_. (**c**) The Fourier infrared spectrograms of the catalysts of BiVO_4_, CuBi_2_O_4_, 0.2-CuBi_2_O_4_/BiVO_4_, 0.5-CuBi_2_O_4_/BiVO_4_, and 0.8-CuBi_2_O_4_/BiVO_4_. (**d**) SEM images of the catalysts of 0.5-CuBi_2_O_4_/BiVO_4_. (**e**) EDX mapping image of 0.5-CuBi_2_O_4_/BiVO_4_. (**f**) HRTEM image of 0.5-CuBi_2_O_4_/BiVO_4_. (**g**) The interface structure between CuBi_2_O_4_ and BiVO_4_ simulated.

**Figure 2 nanomaterials-15-00471-f002:**
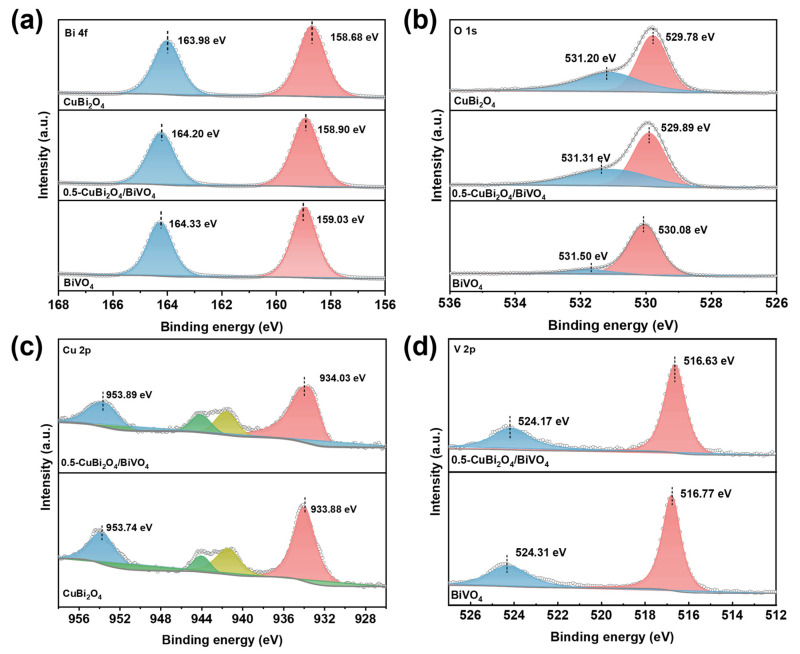
The XPS spectra of (**a**) Bi 4f, (**b**) O 1s, (**c**) Cu 2p, and (**d**) V 2p of CuBi_2_O_4_, BiVO_4_, and 0.5-CuBi_2_O_4_/BiVO_4_.

**Figure 3 nanomaterials-15-00471-f003:**
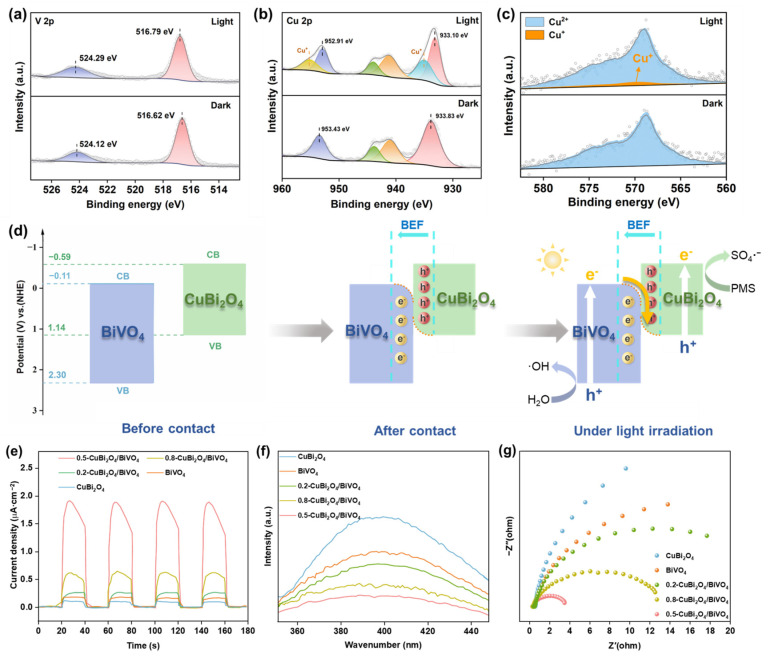
The in situ XPS spectra of (**a**) V 2p and (**b**) Cu 2p and the in situ AES spectra of (**c**) Cu 2p under light and dark conditions. (**d**) The charge transfer mechanism of 0.5-CuBi_2_O_4_/BiVO_4_ under illumination. The (**e**) transient photocurrent curve, (**f**) PL spectrogram, and (**g**) EIS diagram of the catalyst.

**Figure 4 nanomaterials-15-00471-f004:**
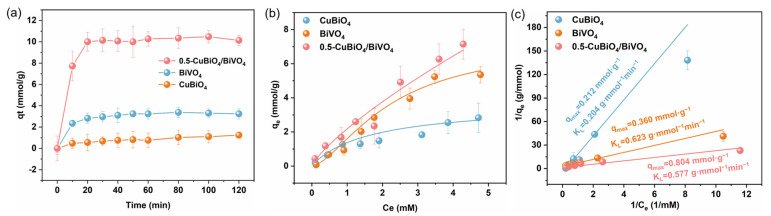
(**a**) Adsorption capacities of 0.5-CuBi_2_O_4_/BiVO_4_, BiVO_4_, and CuBi_2_O_4_ for PMS at different time intervals. (**b**) Influence of the initial concentration on the adsorption of PMS by CuBi_2_O_4_, BiVO_4_, and 0.5-CuBi_2_O_4_/BiVO_4_. (**c**) Langmuir isotherm plots for the adsorption of PMS on CuBi_2_O_4_, BiVO_4_, and 0.5-CuBi_2_O_4_/BiVO_4_.

**Figure 5 nanomaterials-15-00471-f005:**
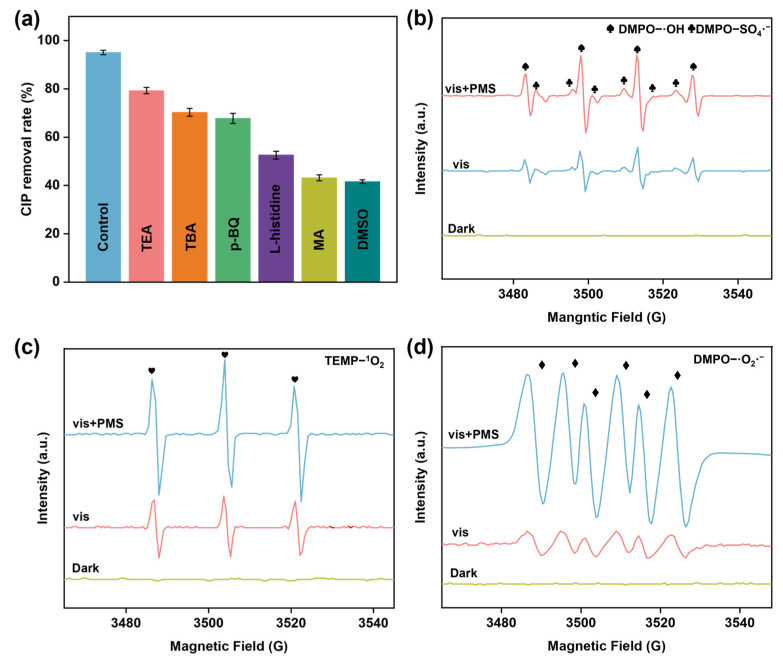
(**a**) Removal of CIP at 30 min after addition of different active substance bursting agents. EPR spectra of (**b**) DMPO-•OH and SO_4_•^−^ in aqueous dispersions. (**c**) TEMP-^1^O_2_ in aqueous dispersion. (**d**) DMPO-•O_2_^−^ in dimethyl sulfoxide dispersions.

**Figure 6 nanomaterials-15-00471-f006:**
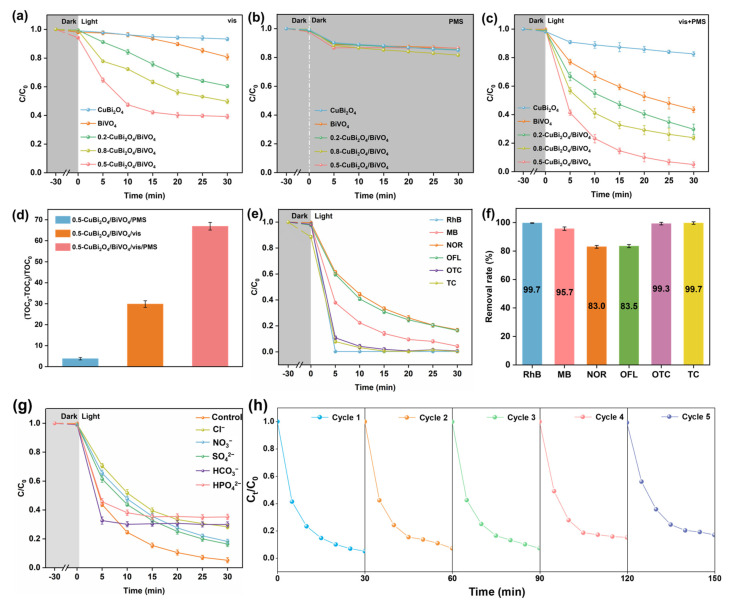
The degradation curves of CIP by different catalysts (**a**) under visible light, (**b**) PMS conditions, and (**c**) visible light + PMS conditions. (**d**) TOC removal rates under different conditions. (**e**) The degradation curves and the removal rate within 30 min of the 0.5-CuBi_2_O_4_/BiVO_4_/vis/PMS system for other antibiotics and (**f**) organic pollutants. The effects of (**g**) other interfering ions on the degradation efficiency of CIP and (**h**) the stability test in 0.5-CuBi_2_O_4_/BiVO_4_/vis/PMS system. The concentrations of catalysts, PMS, all organic pollutants, and pH in (**a**–**f**) are 400 mg/L, 1 mmol/L, 10 mg/L, and 7, respectively.

**Figure 7 nanomaterials-15-00471-f007:**
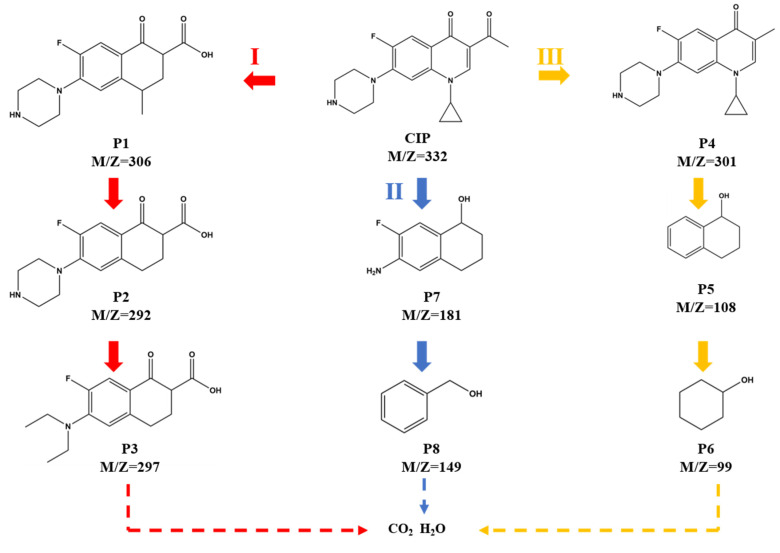
Three pathways of CIP degradation which cotain dehydroxylation, demethylation, defluorination, hydroxylation, and complete rupture of the piperazine ring structure.

## Data Availability

Data are contained within the article.

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
