# Peer review of "Synergistic Photocatalytic Oxidation and Reductive Activation of Peroxymonosulfate by Bi-Based Heterojunction for Highly Efficient Organic Pollutant Degradation"

_nanomaterials, 2025, doi:10.3390/nano15060471_

Round 1
Reviewer 1 Report
Comments and Suggestions for Authors
The authors present an idea of constructing S-type heterojunction by employing CuBi2O4/BiVO4 composite, which exhibited a high efficiency for synergistic peroxymonosulfate (PMS) photoactivation and photocatalytic oxidation of organic pollutants. The built-in electric field in CuBi2O4/BiVO4 composite of S-type heterojunction inhibited the fast recombination of photo-excited electron-hole pairs and enabled a high reduction potential of photo-excited electron to reduce O2 to •O2– and a high oxidation potential of photo-excited hole to oxidize H2O to •OH. This manuscript is overall well presented, the experiments are reasonably designed, and the conclusions are relatively solid. Therefore, I recommend considering the publication of this manuscript in Nanomaterials after minor revisions.
Specific comments are listed below:
- In line 26 and 27 of Abstract, the author claimed that “under visible light irradiation, 95.1% of ciprofloxacin (CIP) can be completely degraded into CO2 and H2O”. However, there is no solid evidence in the paper that 95.1% of CIP was completely degraded to CO2 and H2O, it is more likely that they might just decompose into smaller organic molecules.
- For the XRD spectra in Fig.1b, it seems that the synthesized CuBi2O4 was not quite pure. For example, what is the XRD peak around 13° in the red circle? In addition, compared to the standard PDF#79-1810, the XRD peaks of synthesized CuBi2O4 seem to be right-shifted (e.g., peaks at 2θ = 21.0°, 28.2°)
- In Fig. 1c, please specify the IR peaks of CuBi2O4 around 550 cm-1 and 850 cm-1?
- In Fig.3a - 3c, the authors observed a binding energy of V 2p increasing by 0.17 eV, while a binding energy of Cu 2p decreasing by 0.73 eV for in situ XPS of 0.5-CuBi2O4/BiVO4 by switching from dark to light illumination to demonstrate the interface charge flow. However, for a better comparison, in situ XPS characterizations of pure CuBi2O4 and BiVO4 are recommended to check whether binding energies of V 2p and Cu 2p would change.
- In Figure 6e and 6f, please clarify the amount and concentrations of these organic pollutants that were used during photo-degradation experiments.

Author Response
Comments 1: In line 26 and 27 of Abstract, the author claimed that “under visible light irradiation, 95.1% of ciprofloxacin (CIP) can be completely degraded into CO2 and H2O”. However, there is no solid evidence in the paper that 95.1% of CIP was completely degraded to CO2 and H2O, it is more likely that they might just decompose into smaller organic molecules.
Response 1: Thank you for pointing this out. We are sorry for the unrigorous claim. Therefore, we have made revision to the expression in the manuscript to enhance rigor.
Revision to the main text in line 26, page 1: “Therefore, under visible light irradiation, 95.1% of ciprofloxacin (CIP) can be degraded.”
Comments 2: For the XRD spectra in Fig.1b, it seems that the synthesized CuBi2O4 was not quite pure. For example, what is the XRD peak around 13° in the red circle? In addition, compared to the standard PDF#79-1810, the XRD peaks of synthesized CuBi2O4 seem to be right-shifted (e.g., peaks at 2θ = 21.0°, 28.2°)
Response 2: Thank you for pointing this out. We propose that the peak observed at 13° corresponds to the reflection at approximately 14° in the standard reference pattern. This slight shift (Δ2θ ≈ 1°) is likely attributed to structural defects, such as lattice distortions or oxygen vacancies. In addition, testing temperature and machine errors are likely to be the reasons for the slight shift of characteristic peaks in XRD. Moreover, the intensity ratio of its main characteristic peaks remains consistent with that in the standard card, indicating the successful synthesis of CuBi2O4.
Comments 3: In Fig. 1c, please specify the IR peaks of CuBi2O4 around 550 cm-1 and 850 cm-1?
Response 3: Thank you for pointing this out. According to reported literatures, the peak at 550 cm−1 and 850 cm-1 usually corresponds to the stretching vibration of Cu-O bond and Bi-O bond, respectively.
Revision to the main text in line 151, page 5: “For the original CuBi2O4, the peak at 859 cm-1 and 1398 cm−1 are related to the stretching vibration of the Bi–O bond [26, 27]. The peak at 556 cm-1 is related to the stretching vibration of the Cu–O bond[28].”
Reference: Construction and performance of a novel CuBi2O4/In2O3 Z-scheme heterojunction photocatalyst, Materials Science in Semiconductor Processing, 160, 2023, 107464 https://doi.org/10.1016/j.mssp.2023.107464
Well-constructed CeO2-coated CuBi2O4 heterojunction: Enhanced charge carriers transportation, Surfaces and Interfaces, 59, 2025, 105921
https://doi.org/10.1016/j.surfin.2025.105921
Comments 4: In Fig.3a - 3c, the authors observed a binding energy of V 2p increasing by 0.17 eV, while a binding energy of Cu 2p decreasing by 0.73 eV for in situ XPS of 0.5-CuBi2O4/BiVO4 by switching from dark to light illumination to demonstrate the interface charge flow. However, for a better comparison, in situ XPS characterizations of pure CuBi2O4 and BiVO4 are recommended to check whether binding energies of V 2p and Cu 2p would change.
Response 4: Thank you for pointing this out. According to the reported literatures, single component CuBi2O4 or BiVO4 can undergo electron transfer under light excitation and they can only undergo oxidation or reduction reactions independently during the catalytic process, which can easily cause the recombination of photo generated electron hole pairs. Testing them does not provide effective assistance in analyzing the electron transfer of 0.5-CuBi2O4/BiVO4.Therefore, we believe that conducting in-situ XPS testing on individual components is not necessary.
Reference: Insight into combining visible-light photocatalysis with transformation of dual metal ions for enhancing peroxymonosulfate activation over dibismuth copper oxide, Chemical Engineering Journal, 397, 2020, 125310,
https://doi.org/10.1016/j.cej.2020.125310
The transition of tetragonal to monoclinic phase in BiVO4 coupled with peroxymonosulfate for photocatalytic degradation of tetracycline hydrochloride, Environmental Research, 267, 2025, 120631
https://doi.org/10.1016/j.envres.2024.120631
Comments 5: In Figure 6e and 6f, please clarify the amount and concentrations of these organic pollutants that were used during photo-degradation experiments.
Response 5: Thank you for pointing this out. This was indeed an oversight on our part and we have already supplemented the corresponding organic pollutant concentrations.
Revision to the main text in line 403, page 13: “The concentrations of catalysts, PMS, all organic pollutants and pH in (a-f) are 400 mg/L, 1 mmol/L, 10 mg/L and 7, respectively.”
Reviewer 2 Report
Comments and Suggestions for Authors
I have carefully reviewed the manuscript and acknowledge the significance of this study in advancing photocatalytic and peroxymonosulfate (PMS) activation technologies. The authors have provided a thorough investigation of the S-type heterojunction x-CuBi2O4/BiVO4 system, demonstrating its efficiency in organic pollutant degradation. However, to ensure the robustness and completeness of the presented data, I recommend addressing the following concerns before the manuscript is accepted for publication:
- EIS Analysis and Equivalent Circuit Analysis
The manuscript presents electrochemical impedance spectroscopy (EIS) data to support charge transfer efficiency. However, EIS analysis should be complemented with an equivalent circuit analysis to better interpret charge transfer resistance and interfacial characteristics. A suggested reference for this methodology is Journal of Alloys and Compounds 816, 152513, https://doi.org/10.1016/j.jallcom.2019.152513 (not need to be cited). The authors should incorporate an equivalent circuit model and provide fitted parameters to strengthen their discussion on electron transport mechanisms.
- Adsorption and Desorption Curve Behavior in Figures S6d and S6e
The adsorption and desorption curves in Figures S6d and S6e appear to be crossing, which requires further clarification. The authors should explain whether this is due to capillary condensation effects, structural characteristics of the catalysts, or experimental inconsistencies. Addressing this issue will enhance the credibility of the adsorption study.
- Logarithmic Scale for X-Axis in Figures S6a to S6c
To improve the clarity and readability of the adsorption kinetic data, the x-axis in Figures S6a to S6c should be presented in logarithmic form. This adjustment is commonly used in adsorption studies and will facilitate better interpretation of the data trends.
- CHNS Elemental Analysis of Fresh and Spent Catalysts
The authors should verify the extent of mineralization by conducting CHNS elemental analysis on both fresh and spent catalysts. This additional characterization will help confirm whether the TOC results genuinely reflect complete mineralization or if some degradation products remain adsorbed on the catalyst surface. This analysis is crucial for assessing the long-term stability and reusability of the catalyst system.
Upon addressing these points, the manuscript will be significantly improved and suitable for publication. I appreciate the authors' efforts in advancing photocatalytic technology and look forward to their revisions.
Author Response
Comments 1: The manuscript presents electrochemical impedance spectroscopy (EIS) data to support charge transfer efficiency. However, EIS analysis should be complemented with an equivalent circuit analysis to better interpret charge transfer resistance and interfacial characteristics. A suggested reference for this methodology is Journal of Alloys and Compounds 816, 152513, https://doi.org/10.1016/j.jallcom.2019.152513 (not need to be cited). The authors should incorporate an equivalent circuit model and provide fitted parameters to strengthen their discussion on electron transport mechanisms.
Response 1: Thank you for pointing this out. We agree with this comment. Therefore, we fitted the EIS data and added an equivalent circuit diagram in the Figure S5 and Table S1.
Revision to the main text in line 286, page 9: “Figure S5 illustrates the equivalent circuit diagram employed for the EIS analysis of 0.5-CuBi2O4/BiVO4, and Table S1 summarizes the corresponding fitting parameters.”
Figure S5 EIS-analyses of 0.5-CuBi2O4/BiVO4 using equivalent circuits.
Parameter Error
Resistance 1 7.75
Capacitance 1 13.8
Resistance 1 6.23
Capacitance 2 1.69
Resistance 2 0.69
Capacitance 3 8.36
Resistance 3 14.02
Table S1 The fitted parameters of the equivalent circuits.
Comments 2: The adsorption and desorption curves in Figures S6d and S6e appear to be crossing, which requires further clarification. The authors should explain whether this is due to capillary condensation effects, structural characteristics of the catalysts, or experimental inconsistencies. Addressing this issue will enhance the credibility of the adsorption study.
Response 2: Thank you for your valuable comments. One possible reason is all samples has small surface areas (<10 m2/g), thus possible crossing due to measuring error is plausible. On the other hand, according to the BET pore size distribution analysis, the material is predominantly composed of micropores, accompanied by a minor fraction of mesopores and macropores. The N2 adsorption-desorption isotherms indicate that the adsorption behavior is primarily dominated by microporous adsorption, characterized by the formation of single or few molecular layers. As a composite of metal oxides, 0.5-CuBi2O4/BiVO4 exhibits well-ordered, straight channels of mesopores and macropores on its surface. Given the absence of additional energy barriers during gas adsorption and desorption processes, no significant hysteresis phenomenon is observed in the isotherms.
Revision to the supporting information Figure S6: “Samples has small surface areas (<10 m2/g), thus slight crossing due to measuring error is plausible. Given the absence of additional energy barriers during gas adsorption and desorption processes, no significant hysteresis phenomenon is observed in the isotherms[1-3]. Thus this crossing will not affect the conclusion.”
Reference: A green method for the preparation of nitrogen-doped mesoporous carbon and its application in adsorptive desulfurization, New Journal of Chemistry, 48, 2024, 15802-15809. https://doi.org/10.1039/D4NJ02577F
An efficient one-step condensation and activation strategy to synthesize porous carbons with optimal micropore sizes for highly selective CO2 adsorption, Nanoscale, 6, 2014, 4148-4156. https://doi.org/10.1039/C3NR05825E
Synthesis of colloidal silica nanoparticles of a tunable mesopore size and their application to the adsorption of biomolecules, Journal of Colloid and Interface Science, 349, 2010, 173-180
https://doi.org/10.1016/j.jcis.2010.05.041
Comments 3: To improve the clarity and readability of the adsorption kinetic data, the x-axis in Figures S6a to S6c should be presented in logarithmic form. This adjustment is commonly used in adsorption studies and will facilitate better interpretation of the data trends.
Response 3: Thank you for your suggestion. We sincerely appreciate your constructive feedback. In response, we have revised the X-axis of Figure S7a-c to a logarithmic scale and adjusted the coordinate intervals to a discrete format. This modification enhances the readability of the figures and facilitates more precise data interpretation. We believe these adjustments will improve the clarity and accessibility of the presented results.
Figure S7 (a) Pore size distribution diagram and (d) Nâ‚‚ adsorption-desorption curve of CuBi2O4. (b) Pore size distribution diagram and (e) Nâ‚‚ adsorption-desorption curve of BiVOâ‚„. (c) Pore size distribution diagram and (f) Nâ‚‚ adsorption-desorption curve of 0.5- CuBi2O4/BiVO4.
Comments 4: The authors should verify the extent of mineralization by conducting CHNS elemental analysis on both fresh and spent catalysts. This additional characterization will help confirm whether the TOC results genuinely reflect complete mineralization or if some degradation products remain adsorbed on the catalyst surface. This analysis is crucial for assessing the long-term stability and reusability of the catalyst system.
Response 4: Thank you for pointing this out. Although we acknowledge the significance of elemental analysis for the used 0.5-CuBi2O4/BiVO4, the high metal content of the material does not meet the testing conditions by an elemental analyzer. Consequently, we employed Raman spectroscopy to characterize the catalyst and reaction materials after use. The Raman spectra revealed the absence of characteristic peaks corresponding to peroxymonosulfate (PMS) and ciprofloxacin (CIP) on the used catalyst, indicating that no adsorption of CIP occurred on the catalyst surface. This finding further confirms the total organic carbon (TOC) test results.
Figure S13 The Raman spectra of fresh/used 0.5-CuBi2O4/BiVO4, CIP and PMS.
Revision to the main text in line 475, page 15: “In addition, we performed Raman spectroscopy analysis on the used 0.5-CuBi2O4/BiVO4 without washing to investigate potential residual species. As illustrated in Figure S13, no characteristic peaks corresponding to CIP or PMS were detected on the catalyst surface after the reaction. This observation indicates that the residual CIP did not adsorb onto the 0.5-CuBi2O4/BiVO4, further supporting the reliability and accuracy of the TOC test results. Moreover, compared to fresh catalysts, the characteristic peaks did not change, which also indicates good catalytic stability.”
